# Measuring the Contribution of the Bioeconomy: The Case of Colombia and Antioquia

**Mauricio Alviar [1], Andrés García-Suaza [1,2,*] [ID], Laura Ramírez-Gómez [1,3] and Simón Villegas-Velásquez [1,4]**

1 School of Economics and Management, Universidad EIA, Envigado 055428, Colombia; mauricio.alviar@eia.edu.co (M.A.); laura.ramirez5@udea.edu.co (L.R.-G.); svillegasv@unal.edu.co (S.V.-V.)
2 School of Economics, Universidad del Rosario, Bogotá 111711, Colombia
3 School of Economic Sciences, Universidad de Antioquia, Medellín 050010, Colombia
4 School of Sciences, Universidad Nacional de Colombia, Medellín 050041, Colombia
* Correspondence: andres.garcia@urosario.edu.co

**Abstract:** This paper proposes a set of five indicators to monitor the bioeconomy in Colombia and Antioquia, one of the main regions of the country. The proposed indicators encompass the dimensions of sustainability and emphasize the role of knowledge and scientific research as driving forces of the bioeconomy strategies. To estimate the contribution of the bioeconomy to value added, employment, and greenhouse gas emissions, an input–output analysis is carried out. In addition, text mining analysis techniques are implemented to identify the research groups with an agenda related to bioeconomy fields. Our results reveal an important slot to foster the growth of a sustainable bioeconomy that enables local economies to achieve inclusive growth.

**Keywords:** knowledge-based bioeconomy; input-output analysis; sustainable development; bio-based industries; local strategies; Colombia





## 1. Introduction

Society is facing global challenges that require urgent solutions to ensure social, economic, and ecological sustainability. Besides the current economic and social crisis raised by the outbreak of Covid-19, the expected population growth in 2050, the incidence of poverty, the climate change, the over-exploitation of natural resources, among other situations, obligate us to rethink the production model and the drivers of economic growth and social development, which have been based on fossil-fuel resources [1,2]. Therefore, it is imperative to use scientific knowledge and technological progress to seek production models that efficiently exploit and preserve natural resources. Accordingly, many countries have launched different strategies, such as green growth and circular economy as alternatives to pursuit sustainable, resilient, and inclusive economic growth. In this context, both policy and research debates coincide, in that in order to transit towards the green economy the incorporation of sustainable use of biodiversity and the development and strengthening knowledge-based value chains under the concept of bioeconomy is needed.

Bioeconomy has gained particular attention in recent years in the public policy agenda in several countries as an alternative to achieve the 2030 Agenda, its Sustainable Development Goals, and the Paris Agreement through the conversion of bio-based renewable resources into fibers, food, fuels, and chemicals. This has increased the interest in the development of a knowledge-based bioeconomy, mainly directed towards biotechnological advancement. However, bioeconomy is not necessarily a synonym for sustainability since there have been unintentional environmental and social impacts associated with it. In this context, several authors have analyzed the bioeconomy role in sustainability. Loiseau et al. [3] identified that the traditional concept of bioeconomy is more oriented towards a weak approach, since it is focused on the use of natural resources as inputs to productive processes. D'Amato et al. [4] found that the bioeconomy-related policies have been more

oriented towards economic growth, without considering in the same depth the environmental and social pillars of sustainability; a conclusion also reached by Ramcilovic-Suominen and Pülzl [5].

Although bioeconomy is closely related to bio-based economy, green economy, and circular economy, academic literature has identified particularities in the use of the term, which has been linked to the use of biomass and knowledge across different disciplines. In this manner, the bioeconomy implies a transformative change under three visions: biotechnology, bioecology, and bioresources [1]. This imposes the challenge of defining the scope of bioeconomy as well as sets up a framework to measure its development [4,6]. Monitoring the bioeconomy requires to consider two relevant ingredients of the bioeconomy: first, the sustainability perspective, which involves three dimensions: economy, society, and environment, and second, the use of biological knowledge, in particular the capacity of generating and implementing biotechnology, as a driving force to develop the bioeconomy [7–9]. Providing relevant indicators in this regard is crucial not only to monitor the progress of national bioeconomy strategies but also to inform policymakers, society, and organizations about the impact of policies and mapping opportunities to foster communities and businesses.

This paper aims to build a set of indicators aligned with the different dimensions of sustainability and the driving forces of the bioeconomy for Colombia and a representative region, Antioquia. In particular, we estimate (i) the contribution of the bioeconomy to the total value added by using intersectoral biomass flows, (ii) the employment share of bioeconomy-related activities, and (iii) the bioeconomy contribution to greenhouse gas (GHG) emissions, which account for the economic, social, and environmental dimension, respectively [10]. These indicators are compatible with those proposed by the European Commission through the project Biomonitor, which aims to reduce the measurement gaps facilitating the international comparison [11]. In addition, we propose a measurement of the capacity of generating and applying specific knowledge that catalyzes the bioeconomy. To do so, we consider two additional indicators: (i) the enrollment in academic programs related to biological knowledge and technology and innovation, and (ii) the number of research groups working on bioeconomy related fields.

These five leading indicators provide a baseline monitoring framework of the bioeconomy at a national and regional level and the methodological approach can be extended to other regions in Colombia and other countries in Latin America and the Caribbean. Colombia is an interesting case of study because it is one of the 17 megadiverse countries defined by the United Nations Environmental Program. Besides, Colombian governments have made significant efforts to develop and launch bioeconomy policies [12,13], which go hand-in-hand with the national strategies of green growth [14]. Accordingly, an International Mission of Experts aiming to orientate the country's public policy, presents the bioeconomy as one of the strategic focuses for the advancement of science, technology and innovation and has made recommendations to increase the contribution of the bioeconomy to the economy as a whole [15]. In turn, Antioquia is the second most important regional economy after the capital Bogotá. This region is also one of the richest in biodiversity, e.g., shelters 47% of the total species reported in Colombia [16]. The local government has also started designing a route map supported by the Scientific Committee on Climate Change.

This paper proceeds as follows: Section 2 presents a discussion about the concept of bioeconomy and the related literature on its monitoring. Section 3 describes the methodology and data used to compute the proposed indicators. Section 4 shows the results and discussion. Section 5 presents the concluding remarks.

## 2. The Concept of Bioeconomy and Related Literature

From its origins, the term bioeconomy has evolved in the view of the policymakers and academic research, both recognizing the remarkable role of biotechnology as an enabling factor [7]. Birner [17] documents the origin of the term and argues that during the 1960s and 1970s, authors such as Zeman and Roegen use the term "bioeconomics" to refer to

the discussion about the incompatibility between the unlimited economic growth and the availability of natural resources, while the term "bioeconomy" was defined by the geneticists Juan Enriquez and Rodrigo Martinez [18]. Later, Enriquez, in his work entitled "Genomics and the World's Economy", quoted the term and discussed the role of genetic science to transform the industry [19]. Although the concept of bioeconomy has been used in the policy agenda and in the scientific arena from the seventies, it became more important in the last two decades [2,20], especially when the European Commission launched the Bioeconomy Strategy in 2012, motivating national actions to establish priorities and policies to scale the bio-based sectors and to deploy local economies [21,22].

Nowadays, there is a broad consensus about what bioeconomy is, and in fact, the definition provided by the Global Bioeconomy Summit in 2018 has been adopted in many scenarios. According to this definition, bioeconomy refers to "the production, use and conservation of biological resources, including knowledge, science, technology and related innovation, for the provision of information, products and services through all economic sectors towards a more sustainable economic system" [8]. This definition makes clear the role of science, technology, and innovation to move from a production factor-oriented economy towards a more knowledge-oriented economy [2,23]. This scope is in the labeled "knowledge-based bioeconomy", which was aligned with the EU innovation policy perspective under the view of boosting the economic growth by means of high-technology industries, fostering national and regional innovation systems through the rise of the research and development (R&D) expenditure and the specialized labor demand.

Concepts such as green economy, bio-based economy, and circular economy are quite close but have some particularities worthy to mention. Green economy is about ecosystems functioning, low carbon type of economic activities, and even social inclusion. This concept refers to environmental risks and the scarcity signals from the ecological systems. On the other hand, the bio-based economy concept is the closest term to bioeconomy since it is defined as the production of goods and services from biological material or resources. Examples of typical bio-based products are biopolymers and bioplastics. At the same time, the bioeconomy is seen as part of the global economic growth and in that sense closer to the innovation and development perspective. Finally, circular economy refers to a way of producing goods and services recycling and reducing original raw materials including biological resources [24,25].

As delimiting the scope of the bioeconomy is a challenging task, building monitoring indicators aligned with a policy's objective is essential to follow-up the short-term changes in the bioeconomy as well as support the design of long-term policies and programs. There is a growing literature defining relevant indicators to monitor and evaluate the impacts of the bioeconomy, which are focused on the three dimensions of sustainability and encompassed the measurement of the contribution of the bioeconomy to the production, consumption, and employment, the availability and use of natural resources, climate footprint, among others [4,25,26]. However, there is not a unique framework that standardizes the measurement method, a desirable feature to monitor the bioeconomy and facilitate international and inter-regional comparisons [27,28].

In this scenario, an important question for policymakers and organizations is how relevant the bioeconomy is for the whole economy or particular sectors of a country or region. There is a branch of literature devoted to linking the scope of the definition of bioeconomy with the available data on economic activities. Two approaches based on measuring the bio-share content of products and industries have been considered. The first, proposed by The Joint Research Centre (JRC) of the European Commission in collaboration with the Nova-Institute [10,29,30], is based on the views of experts, who are interviewed about the biomass content of products at a high disaggregation level.

The second method is also an input-oriented approach that exploits available information in the input–output (IO) matrices from the System of National Accounts to infer how the different sectors use biological resources from the primary sector (agriculture, forestry, fishery, and aquaculture) as a proxy of the biomass flows under the assumption

that bio-based share of outputs is the same as that of inputs and constant return to scale production technology [31]. Accordingly, bio-shares at the industry level (e.g., using the Nomenclature statistique des activités économiques dans la Communauté européenne—NACE—or the International Standard Industrial Classification—ISIC) are estimated, which enable to measure the contribution of bioeconomy to the gross value added and employment, GHG emissions, and other relevant variables to which sectoral level data is available. The main advantage of this approach is that it can be implemented across countries and regions where IO matrices are available. In fact, it has been implemented for many countries, for instance, Efken et al. [11] for Germany, Bracco et al. [2] for Germany, Argentina, Malaysia, Netherlands, South Africa, and the US, Wen et al. [31] for Japan, Loizou et al. [32] for Poland, and Kuosmanen et al. [10] for EU-28, among others.

Besides the assessment of the impact of the bioeconomy on the dimensions of sustainability, monitoring the driving forces is also vital to get a comprehensive landscape. Driving forces include advances in biological sciences, fostering linkages between supply chain policies and strategies, accelerating the adaptation to climate change, among others enabling factors [25,27]. The knowledge generation capacity in the bioeconomy domain is of special interest since public policy has an active role in promoting cooperation among academic institutions and private organizations to expand the understanding of the local resources and biodiversity, improve the scientific know-how and amplify the market opportunities to develop new byproducts and co-products. Existing literature studying this factor has focused on the case of European countries and quantifies the funding resources, the number of research networks and technology platforms, research projects, and research groups involved in biotechnology-related research activities and topics such as food, agriculture, forestry, fisheries, and bioenergy [1,8,33,34].

The ratio between R&D expenditures and gross domestic product (GDP) has been considered as a leading measure of the capacity of generating knowledge of an economy [35]. In Colombia, this indicator is nearly 0.28% for 2019, whereas the average expenditure on R&D in the OECD countries corresponded to 2.4% [36]. This lack of investment has been outlined in several documents related to the Bioeconomy in Latin America and Colombia as one of the main reasons that hinder its development. However, despite the low R&D investment at national level, Antioquia is the region that has a larger share, corresponding to 31.6% of the national total for the 2017–2019 period [37]. Alternatively to the R&D expenditure, Jaffe et al. [38] proposed that the academic productivity (publications per capita) is more correlated to the GDP and the Human Development Index as measurement of a nation's economic development and welfare than other commonly used variables employed to that end, and, therefore, should be considered in the measurement of economic activities. In South Africa for example, indicators such as full-time equivalent researchers, scientific publications, and bioeconomy-related publications are included within the bioeconomy strategy, to approach the level of scientific development and the contributions of this area of knowledge to the science of the territories [2].

However, these approaches have been concerned with the process of knowledge generation, and the diffusion and use of knowledge are not usually addressed within the analysis of the bioeconomy. For this purpose, Urmetzer et al. [39] proposed to consider three categories of knowledge into the policy agenda of sustainable knowledge-based bioeconomy (SKBB), namely systems knowledge, normative knowledge, and transformative knowledge. The first refers to the understanding of the complexity and interdependence of production, environmental, economic, social, and political processes. The second refers to the desired system state and the evaluation of alternative system states, including not only the economic aspect but also the social and ecological ones. Finally, transformative knowledge refers to the competencies that must be acquired to affect a transgression from the *status quo* to the desired state, accordingly to the other two kinds of knowledge: systems and normative. The transformative knowledge, as its name suggests, could ultimately motivate structural changes to achieve sustainability goals embedded in the bioeconomy [23].

## 3. Methods and Data

### 3.1. Estimation of the Bio-Shares and the Size of the Bioeconomy

To estimate the size of the bioeconomy and its contribution to the three dimensions of sustainability, the bio-based content of the output, named bio-shares, should be estimated. To this end, IO matrices have been proven as a suitable input to measure the flows of biomass and provide granular estimates of the bio-shares at the economic activity level. This approach considers that the economic activities belonging to the primary sector (such as agriculture, forestry, fishery, and aquaculture) are fully bio-based since the whole output is intensive in biological resources. In turn, the remaining economic activities are treated as mixed, assuming that the bio-shares are equivalent to the proportion of inputs from the primary sector (see [10,40], for further details). As the IO matrices are highly standardized across countries, this method allows for international comparisons.

To provide some intuition about the estimation procedure, we consider simplified versions of the IO matrices for two and three sectors (see Table 1). The top part of each IO matrix contains the flow of inputs in monetary units between sectors. For instance, in the case of two sectors, the amount of input denoted by A indicates that Sector I uses inputs equivalent to that value from their own production and uses B from Sector II. In turn, D and E have similar interpretations for the case of Sector II. Each row or column of the IO matrices refers to an economic sector, which are composed of several economic activities. In the bottom part, the IO matrix contains the value added (Vi) and the total output (Oi). In the two sectors case, Sector I and II represent the primary sector (agriculture, forestry, fisheries, and aquaculture) and the rest of the economy (manufacturing and services), respectively. In turn, in the three sectors matrix, Sector II and III correspond to manufacturing and services separately.

**Table 1.** Structure of input–output matrices.

| a. Two Sector IO Matrix | | | b. Three Sector IO Matrix | | | |
|---|---|---|---|---|---|---|
| | Sector I | Sector II | | Sector I | Sector II | Sector III |
| Sector I | A | D | Sector I | A | D | G |
| Sector II | B | E | Sector II | B | E | H |
| | | | Sector III | C | F | I |
| Total inputs | $I_1 = A + B$ | $I_2 = D + E$ | Total inputs | $I_1 = A + B + C$ | $I_2 = D+E+F$ | $I_3 = G + H + I$ |
| Value Added | $V_1$ | $V_2$ | Value Added | $V_1$ | $V_2$ | $V_3$ |
| Output | $O_1 = I_1 + V_1$ | $O_2 = I_2 + V_2$ | Output | $O_1 = I_1 + V_1$ | $O_2 = I_2 + V_2$ | $O_3 = I_3 + V_3$ |

Source: Own elaboration.

Using the structure of the OI matrices, the bio-shares are computed under four different scenarios. The first considers the two sectors matrix, and that economic activities in Sector II have the same average productivity. Therefore, the bio-shares are constant across activities and equivalent to the proportion of inputs coming from the primary sector. The second considers that economic activities differ in bio-based content, and so corresponding specific bio-shares are estimated. The third uses the three sectors matrix and assumes that bio-shares include inputs from the primary sector and a proportion from the others that is equivalent to corresponding bio-based content. Finally, the fourth approach computes the bio-shares as a weighted average of the input and output bio-based content. Below, we describe the details of each scenario.

In the first case, we compute a proportionality factor between inputs and output in Sector II, denoted by $\alpha$, such that $O_2 = \alpha I_2$. That is, $\alpha$ is the output level achieved by each unit of input in Sector II. If all inputs have the same productivity, the bioeconomy output and value added of this sector, given by $O_{2b}$ and $V_{2b}$, respectively, are:

$$O_{2b} = \alpha D \qquad V_{2b} = O_{2b} - D = (\alpha - 1)D \qquad (1)$$

where $D$ represents the total input of Sector II coming from Sector I. Therefore, the value added of the bioeconomy is $V_b = V_1 + V_{2b}$. Noticeable, the bio-shares in Sector II can be computed as $\beta_2 = \frac{D}{I_2}$ that satisfies $V_{2b} = \beta_2(O_2 - I_2)$.

The second approach takes advantage of the heterogeneity of the economic activities belonging to Sector II. Individual industry (or economic activity) bio-shares $\beta_{2i}$ are computed as $\beta_{2i} = \frac{d_i}{I_{2i}}$, where $i$ indexes of the economic activity, $d_i$ is the input that economic activity $i$ uses from Sector I, and $I_{2i}$ is the total input required by economic activity $i$. If we assume that Sector II has $m$. industries, it must be satisfied that $\sum_{i=1}^{m} d_i = D$ and $\sum_{i=1}^{m} I_{2i} = I_2$. Thus, the value added is computed as in the previous case.

The third approach uses the extended version of the IO to three sectors and considers that the bio-shares in Sector II and Sector III include the use of inputs from the primary sector and a proportion from the others that is equivalent to the bio-based content of each industry. Consequently, the bio-share of the economic activity $j$ is given by:

$$\gamma_{3j} = \frac{g_j + \sum_{i=1}^{m} \beta_{2i} h_j^i}{I_{3j}} \tag{2}$$

where $t_r^s$ is a general notation indicating the amount of input from the economic activity $s$ used by $r$, and $t_r$ is the total input demanded by the economic activity $j$ such that $\sum_{s=1}^{m} t_r^s = t_r$. That is, $g_j$ is the total input used by the economic activity $j$ from the primary sector. Therefore, the total value of bioeconomy is $V_b = V_1 + V_{2b} + V_{3b}$ where:

$$V_{2b} = \sum_{i=1}^{l} \gamma_{3i}(O_{2i} - I_{2i}) \qquad\qquad V_{3b} = \sum_{j=1}^{n} \gamma_{3j}(O_{3j} - I_{3j}) \tag{3}$$

where $l$ and $n$ are the number of industries in Sector II and Sector III, respectively, and $m = l + n$.

Finally, the vector $\gamma_{3j}$ represents the bio-based content from the perspective of inputs, however, the fact that the primary sector is fully bio-based corresponds an output perspective. Therefore, to build measure of bio-shares that reconciles both views, additional adjustments are required. In such a way, we construct input bio-shares from the primary sector following Equation (3), and the corresponding bio-based content of output, denoted by $\delta_j$, using the ratio between bioeconomy output and total output of each economic activity. According to the fourth approach, the bio-shares the weighted average of $\gamma_{3j}$ and $\delta_j$, that is:

$$\theta_j = \gamma_{3j}\frac{I_j}{O_j} + \delta_j\frac{V_j}{O_j} \tag{4}$$

The final estimates of bio-shares $\theta_j$ can be used to infer the contribution of the bioeconomy in many other outcomes; for instance, employment and GHG emissions. To compute the bio-shares, we use the IO matrix from the Colombian System of National Accounts, whose information is publicly available for 2017 on the website of the national statistical institute, DANE (Data can be downloaded at https://www.dane.gov.co/index.php/estadisticas-por-tema/cuentas-nacionales/cuentas-nacionales-anuales/matrices-complementarias#matriz-insumo-producto, last accessed on 8 August 2020). The IO matrix encompasses 68 economic activities under the International Standard Industrial Classification (ISIC). The primary sector consists of 5 activities, while manufacturing and services group 34 and 29 activities, respectively. The bio-shares for the Colombian economy are also used to compute the size of the bioeconomy in Antioquia, which is the only region in the country with an equivalent account system at the municipality level. Available information includes the value added for the 125 municipalities for 14 economic activities, which can be mapped trough a crosswalk table aggregating the Colombian IO matrix. Therefore, the estimation of the contribution of the bioeconomy in Antioquia and its municipalities assumes the same bio-shares as of the whole Colombian economy. This assumption is reasonable since the Antioquia's economy bears similarities with the case of Colombia.

### 3.2. Contribution of the Bioeconomy to Employment

The behavior of the labor market is usually monitored using household surveys, which provide monthly data about the socioeconomic characteristics and occupations of individuals. In the case of Colombia, this survey, called *Gran Encuesta Integrada de Hogares* (GEIH), is also available on the DANE website (Data can be downloaded at http://microdatos.dane.gov.co/index.php/catalog/659/get_microdata, last accessed on 8 August 2020). GEIH contains information of about 30 thousand households monthly that allows generating representative measures for the national labor market and at the regional (equivalent to state) level. Among other characteristics, GEIH collects information about the economic activity of the workers, which makes it possible to correlate the employment composition with the estimated bio-shares. We use the wave 2017 to estimate the share of the total employment, $e_b$, in Colombia and Antioquia as follows:

$$e_b = \frac{\sum_{i=1}^{M} \theta_i E_i}{\sum_{i=1}^{M} E_i} \tag{5}$$

with $M$ the total economic activities, $\theta_i$ is the bio-share, and $E_i$ the total employment of the economic activity $i$. Although GEIH reports the economic activity in the same classification as the IO matrix, the construction of a crosswalk table is required as the former considers a more granular disaggregation at 4-digits.

### 3.3. Bioeconomy Contribution to GHG Emissions

GHG emissions are one of the most relevant factors causing the acceleration of climate change. Consequently, providing information about the trends in GHG emission has become an important indicator from the environmental perspective. According to the Kyoto protocol, there are six main GHGs, namely: Carbon dioxide ($CO_2$); Methane ($CH_4$); Nitrous oxide ($N_2O$); Hydrofluorocarbons *(HFC$_s$)*; Perfluorocarbons *(PFC$_s$)*; and Sulfur hexafluoride *(SF$_6$)* [41]. In order to make international comparisons, the GHG emissions have been computed in carbon dioxide equivalents *(CO$_{2eq}$)* [10]. In the case of Colombia, information about total GHG emissions at the economic activity level is available in the project CAIT Climate Data Explorer from the World Resources Institute that can be accessed through the Climate Watch website (data can be downloaded at https://www.climatewatchdata.org/data-explorer/historical-emissions?historical-emissions-data-sources=cait&historical-emissions-gases=all-ghg&historical-emissions-regions=All%20Selected&historical-emissions-sectors=total-including-lucf&page=1, last accessed on 12 January 2021). This project collects historical data on GHG emissions for the main Intergovernmental Panel on Climate Change (IPCC) sectors, which can be mapped into seven aggregated economic activities (the seven economic activities are: Agriculture, Forestry, Manufacturing and Construction, Electricity and Heat, Waste, Transportation, and Other sectors) that are compatible with the ISIC. As in the case of employment, a crosswalk table between the IO matrix and the GHG emissions is built. In this way, using the bio-shares for these seven activities, the bioeconomy contribution to GHG emission is computed as the ratio between the proportion of the emissions due to the bioeconomy and the total. (An alternative approach is based footprint calculations. The footprints usually employed are Agricultural Land, associated with land use change, Forest, Water, Material, and Climate [29]. However, the availability of information at a local level is often scarce). The assumption behind this is that the composition of GHG emission within economic activities is equivalent to the bio-shares.

In the case of Antioquia, estimates of the GHG emissions are not available. Therefore, we compute a coefficient that measures the intensity in the $CO_2$ equivalent emissions per unit of value added for the seven economic estimates, given by $\frac{GHG_i}{V_i}$, where $GHG_i$ and $V_i$ are the emissions and the value added in the economic activity $i$. That is, we assume that the average production technology in Colombia is comparable with that in Antioquia. These ratios are used to compute the emissions at the economic activity level defined as $V_{ji}\frac{GHG_i}{V_i}$, with $V_{ji}$ representing the value added of the region $j$ (either Antioquia or any

or its municipalities) in the economic activity *i*. Accordingly, the bio-shares are used to compute the contribution of the bioeconomy to GHG emissions for the regional level $ghg_{jb}$ as follows:

$$GHG_{ji} = V_{ji}\frac{GHG_i}{V_i}, \qquad ghg_{jb} = \frac{\sum_{i=1}^{M_j} \delta_i GHG_{ji}}{\sum_{i=1}^{M} GHG_{ji}} \tag{6}$$

*3.4. Knowledge and Research Capacities of the Bioeconomy*

To study the level of bioeconomy-relevant transformative knowledge within the policy agenda in Colombia or Antioquia is a difficult task. Since biological knowledge has been identified as one of the main driving forces of a bioeconomy strategy, two indicators that approximate the knowledge relevant to the bioeconomy in Colombia and Antioquia are proposed. In particular, the first is the total enrollment in higher education in academic programs that are directly related to the bioeconomy, and the second is a proxy of research capabilities, which is computed by analyzing the research fields and publications of research groups recognized by the Colombian Science Ministry. The intuition behind these indicators is that the supply of human capital and academic research production allows mapping the installed capacities in the region to undertake the transformation toward sustainable development. Nevertheless, these indicators also provide insights into the gaps that might be prioritized by local authorities.

With respect to enrollment in higher education, we use data from the National Higher Education Information System (SNIES, in Spanish) for 2018, which contains the number of students enrolled at the different academic programs including the undergraduate and graduate levels. We defined the following four areas as bioeconomy related programs: engineering and related (agricultural, forestry, agro-industrial, food, and related, agronomic, environmental, sanitary, biomedical, and chemical engineering), agronomy, veterinary medicine, and related (agronomy, veterinary medicine and zootechnics), mathematics and natural sciences (biology, microbiology and related, and chemistry and related), and health sciences (bacteriology and public health). The selection of these areas and academic programs was made considering the adopted definition of bioeconomy, where technological transformation, biological knowledge, and the use of biomass are pillars. Therefore, one of the criteria to select these programs is to see how they are oriented to transforming production processes in search of better practices and less environmental damage, or the creation of new biological knowledge. Consequently, the indicator of human capital training is defined as the enrollment in the selected area to the total by training level (undergraduate and graduate level).

To identify the research capabilities in bioeconomy-related fields in Colombia and the region of Antioquia, we use the information for all the research groups (RG) in Colombia and Antioquia registered on the *Scienti* (*Scienti* is a web platform created by the Colombian Ministry of Science and Technology to facilitate the management of information of the National System of Science and Technology, https://minciencias.gov.co/scienti, last accessed 2 November 2020) portal of the Ministry of Science, Technology, and Innovation. *Scienti* encompasses different information repositories containing detailed information on RG and researchers, respectively. Data processing is carried out using R statistical software and several R packages to implement web scrapping and text mining techniques along with automated language detection [42–46].

We focus our analysis on the repository called GrupLAC that collects specific information of the RG, e.g., RG identifiers, RG name and vision, research fields, and the list of publications and location. RG are classified among 13 National Programs of Science and Technology (NPST), named: Environment, Biodiversity and Habitat; Biotechnology; Basic Science; Agricultural Science; Social Science, Humanities and *Education; Energy and Mining, Geosciences; Engineering; Water Resources; Health Science; Security and Defense; Information and Communication Technology (ICT);* and *Uncategorized*. We conducted three systematic searches to identify the bioeconomy-related research groups (BRG). First, we identified those RG that included explicitly the word "bioeconomy" in their research fields.

Second, we searched for RG that included words beginning with the prefix "bio" (ˆbio*), "sustainable development", or "sustainability" to have a broader scope of groups working in bio-based knowledge. Third, we searched for a list of specific terms related to bioeconomy and biotechnology, which might be a proxy of the potential to generate research and innovation to boost the bioeconomy regional strategy (Table 2). Therefore, the indicator of research capacity is defined as the share between BRG and the total RG in Colombia and Antioquia.

**Table 2.** Terms related to bioeconomy and biotechnology used as search criteria for research groups (RG).

| Search Criteria | | | |
|---|---|---|---|
| ˆbio * | Bioenergy | Ecology | GMO |
| agricultural | biofuel | ecosystem | microorganisms |
| agriculture | biomass | ecosystem service | plant |
| agroindustrial residues | biophysics | ecosystem services | rhizosphere |
| algae | biorefineries | ecosystems | second generation biofuels |
| animal | biorefinery | first generation biofuels | sustainability |
| biocatalysis | biotechnology | fungi | sustainable development |
| biodiesel | botanic | forest | third generation biofuels |
| biodiversity | cell culture | genetic | vegetal |
| bioeconomy | crops | genetically modified organisms genomics | zoology |

Source: own elaboration. Note: ˆbio* means take words beginning with the prefix "bio".

To assess research trends associated with the national bioeconomy, an additional exploratory analysis consisting of identifying the most recurrent words from the publications' titles of the BRG is performed. Specifically, it is proposed to analyze the top ten words of the BRG belonging to each of the NPST that matched the search criteria. The words are selected after adapting the publication's title language, which are mainly Spanish and English.

## 4. Results and Discussion

### 4.1. Indicator 1: Contribution of the Bioeconomy to the Value Added

Using the IO matrix for 2017, the contribution of the bioeconomy to the value added and the GDP is computed implementing the four alternative methods of the bio-shares and presented in Table 3. By definition, 100% of the Sector I output is considered as bioeconomy, while important differences in Sector II and III are revealed across the methods. The estimates suggest that contribution of bioeconomy in manufacturing and services might vary from 7% up to 19.15% (the latter is the summation of the two sectors according to method 4). Using the fourth approach as benchmark, the contribution of the bioeconomy in the value added is 13.71% and 7.25%.

**Table 3.** Bioeconomy contribution to value added and GDP by sector.

| | Method 1 | Method 2 | Method 3 | Method 4 |
|---|---|---|---|---|
| Sector I | 100.00% | 100.00% | 100.00% | 100.00% |
| Sector II | 7.01% | 2.64% | 9.24% | 14.97% |
| Sector III | | | 2.84% | 4.18% |
| % VA | 13.55% | 9.49% | 11.33% | 13.71% |
| % GDP | 7.17% | 5.02% | 5.99% | 7.25% |

Source: Own calculations

The bioeconomy at the economic activity level can be estimated. Figure 1 shows the composition of the bioeconomy considering a disaggregated version of Sector I. Results suggest that the primary sector accounts for 52% of the bioeconomy, while the manufacturing sector represents 28% and the services sector approximately 20%. By decomposing the contribution of each economic activity in Sector I, it is observed that agriculture represents 30% of the total bioeconomy, following that, the cattle raising (13%) and coffee (6%). The latter is one of the main commodities of the Colombian economy. These findings reveal the relevance of agriculture to foster rural development through the introduction of good practices and sustainable management of resources, which would not only improve the working conditions of farmers but could also generate a higher added value.

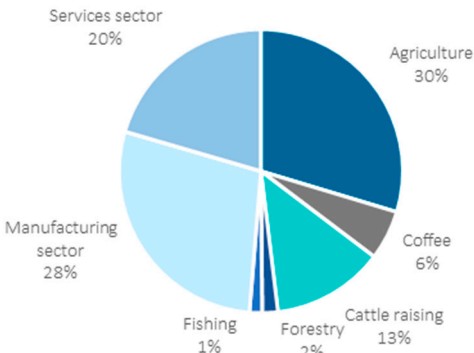

**Figure 1.** Participation of economic sectors in the bioeconomy. Source: Own calculations.

Analyzing the composition of the bioeconomy within the manufacturing and services, estimates show that services related to food and beverages (20% of the bioeconomy excluding the primary sector) and the production of meat products (10%) represent the highest participation of the bioeconomy (see Figure 2). Interestingly, these activities are closely related to agroindustry and provide a natural link between Sector I and the rest of the supply chain. Indeed, projects aiming to implement circular economy models have been developed in these industries. Figure 2 also shows other economic activities for which the participation in the bioeconomy is 2% or higher, some of them refer to foodstuff.

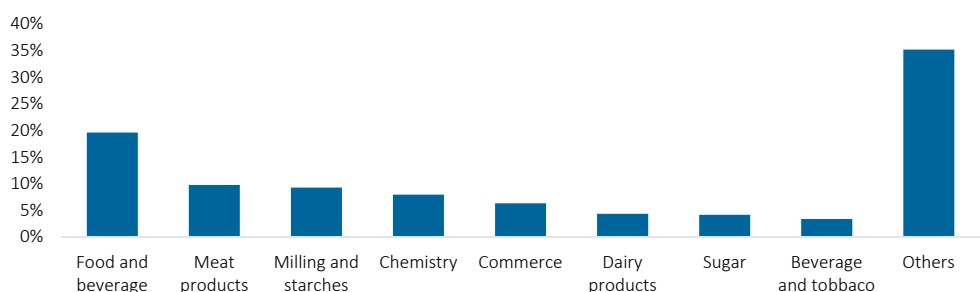

**Figure 2.** Main activities in the manufacturing and services sectors. Source: Own calculations.

The bio-shares of these economic activities are also informative about the importance of the bioeconomy. Regarding the estimates of the bio-shares, important variations are observed across the economic activities. For instance, bio-shares are higher than 80% for coffee products, meat products, and sugar. Other activities with bio-shares higher than 50% are dairy products, millings and starches, and oils.

Overall, the contribution of the bioeconomy in Colombia is similar to the average for EU-28 found by Kuosmanen et al. [10]. However, the composition of the value added differs as the primary sector in Colombia accounts for most of the bioeconomy value added, while this sector exhibits the lower participation for EU-28. In the case of Colombia, the relevance of the primary sector makes it important to solve the puzzle of low agriculture productivity by promoting technology and bio-based knowledge, which is also a challenge in the

global south. It is also important to encourage the transformation of manufacturing and services, which might complement the high potential given by the endowment of natural resources in the region. Therefore, defining priorities associated with the structure of national economies is crucial for the success of bioeconomy strategies.

Bioeconomy provides important opportunities to local economies to foster sustainable development from the social, economic, and environmental perspective taking into consideration the comparative advantages of the regions. We analyzed the contribution of bioeconomy for the region of Antioquia, which is the second regional economy in Colombia and it is endowed with an important stock of biodiversity and natural resources. Antioquia is an interesting case given that it has agricultural production but it has also a mature manufacturing industry with a remarkable participation of agroindustry. Antioquia is located in the northwest of the country and is divided into 125 municipalities, organized into nine subregions. According to the most recent population census, Antioquia has 6.6 million inhabitants (13.5% of the Colombia population), and 4 million living in the metropolitan area that group 10 municipalities including Medellín, the capital city. Antioquia's economy accounts for 14.5% of the Colombian GDP, at the same time the contribution to the manufacturing sector is equivalent to 19%. Mining and financial services are also important economic activities for the regional economy.

Although the economic account system is limited at the regional level, Antioquia is the only region in Colombia with the available information for 14 economic activities at each municipality. Estimating bio-shares at the national level for this aggregated version of the economic activities, we exploited the heterogeneity in the economic structure of the municipalities. In this manner, municipalities specialized in agriculture would report a higher participation in the bioeconomy, and similarly it would be observed for those with an important participation in agroindustry. The contribution of the bioeconomy to the value added in Antioquia is 11.2%, a percentage that is similar, but lower than that reported by Colombia. According to participation by regions, Figure 3 presents the size of the bioeconomy for the nine regions of Antioquia. Remarkably, there are regions where bioeconomy contributes up to half of the total value added. This is the case of the Suroeste and Urabá regions, whose economies are based on agriculture (coffee and banana, mainly) and tourism.

In addition, Figure 3 presents the estimates for all sectors, but also the contribution of the bioeconomy excluding Sector I (see panel b). This analysis allows us to identify the regions with a relevant primary sector, but also those with a higher linked value chain, which is the case of Oriente and Norte. Regarding the results at the municipality level, our results show that the bioeconomy accounts for more than 40% in 14 municipalities. A considerable number of these municipalities are in the southwest, which is a region traditionally associated with agriculture and coffee production, and the north where the dairy industry is one of the main economic activities. We also estimate how municipalities participate in the total of the bioeconomy in Antioquia. These estimates show that Suroeste, Norte, and Urabá (Urabá is a zone specialized in the production of bananas, one of the main export products in Colombia) are the regions with higher participation with 25%, 23%, and 22%, respectively. These findings show the importance of developing local bioeconomy agendas as a response to the specific sectoral composition. This challenge has been also identified in the Colombian National Strategy of Green Growth [14], which asserts that two crucial issues to foster local economies sustainable development in Colombia are the low productivity of agriculture and the absence of public goods and infrastructure that facilitate the access to the markets.

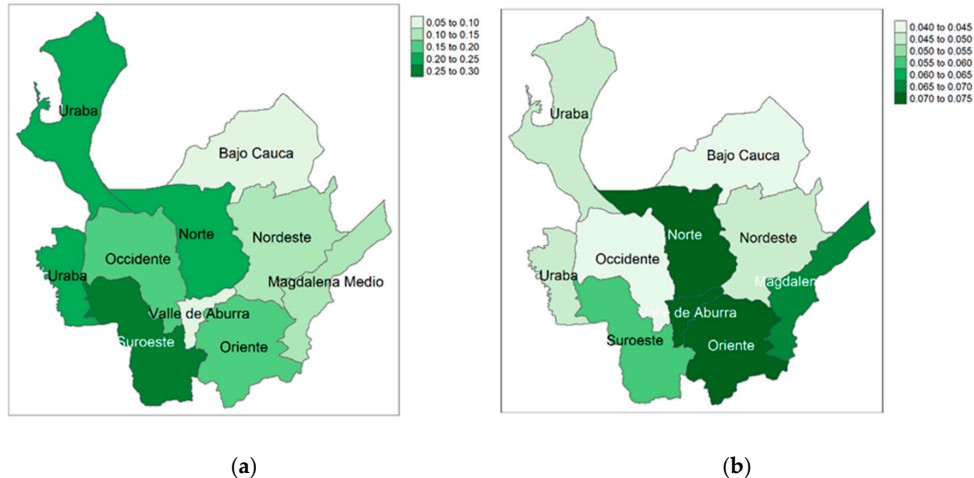

(**a**)  (**b**)

**Figure 3.** Contribution of bioeconomy to value added in Antioquia at subregion level. Source: Own calculations. (**a**) All sectors; (**b**) Excluding Sector I.

*4.2. Indicator 2: Contribution of the Bioeconomy to the Employment*

Considering the estimated bio-shares and a crosswalk table between the IO matrix and the GEIH, we estimate the number of jobs that can be associated with bioeconomy through the interaction between the bio-shares and the distribution of workers across activities. Our results reveal that the contribution of the bioeconomy to employment in Colombia is 4 million jobs, which is 18.2% of the total in 2017. Decomposing the participation of the three sectors, we observed that the primary sector accounts for 70.7%, higher than the participation in value added. This result might be related to the challenge that agricultural productivity is one of the main bottlenecks of the development of rural areas in Colombia. In turn, the participation of manufacturing and services are 11.5% and 17.6%, respectively. In the analysis within sectors, 77.4% of the employment in Sector I is related to the bioeconomy, while in the manufacturing sector is approximately 11.4% and, in the services sector, it is 4.9% (see Figure 4a).

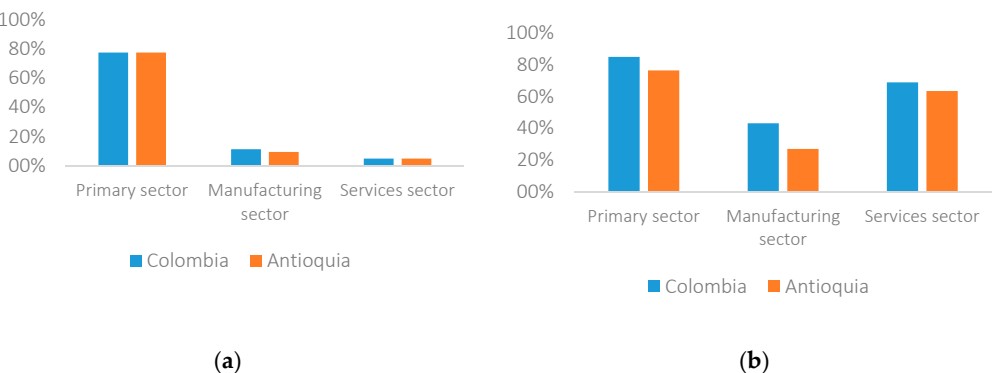

(**a**)  (**b**)

**Figure 4.** Contribution of bioeconomy to employment by sector and informality condition. Source: Own calculations. (**a**) Employment by bioeconomy; (**b**) Informality rate of bioeconomy employment.

On the other hand, the incidence of labor informality in the sectors is analyzed to assess whether there is a relation between bioeconomy and job quality. Figure 4 (panel b) shows informality is higher among bioeconomy employment with respect to the total employment as the primary sector exhibits higher informality rates, close to 85%. In the manufacturing and services sectors, informality rates are 43% and 68.9%, respectively. The latter is also higher than the average in Colombia, which is close to 50%. This result shows the need of designing policies to revert the trend and provide job opportunities under conditions that make it possible to achieve the transition to sustainable production [47].

Filtering the GEIH to compute the employment indicator for Antioquia, we obtain that the contribution of the bioeconomy in the region is 16.4%, 2 p.p. lower than in Colombia. While the contribution of the bioeconomy in the value added is similar, there is a significant difference in both, employment, and informality. These results might be driven by the fact that manufacturing has a higher impact on the bioeconomy in Antioquia compared with the average in Colombia, and that this sector has higher labor productivity and a lower informality rate than the primary sector and services (see Figure 4).

As part of a strategy to accelerate the transition to a green economy, the labor market plays an important role in providing the human capital that enables the transformation of the productive structure. This has been recognized by The International Labour Organization (ILO) that has dynamized the public dialog around the concept of green jobs, which refers to occupations with an identifiable environmental focus. That is, green jobs are defined as occupations that promote environmental protection preservation, economic development, and social inclusion [48]. The Colombian green growth policy has identified opportunities of developing emerging occupations that support the growth of the bioeconomy.

### 4.3. Indicator 3: Bioeconomy Contributions to GHG Emissions

The environmental dimension of sustainability implies a challenge in terms of measurement. Liobikiene et al. [49] identified two main tendencies in the literature regarding the environmental dimension of the bioeconomy. The first approach evaluates the environmental impacts associated with the development of bioeconomy. Such an approach was implemented by Kuosmanen et al. [10] to quantify environmental impacts of GHG emissions in the European bioeconomy. The second approach quantifies the biomass consumption. In this section, we examine the results of the estimates of GHG emissions for the bioeconomy in the case of Colombia and Antioquia.

According to CAIT Climate Data Explorer, GHG emissions in Colombia during 2017 achieved levels of 216.67 $CO_{2eq}$ million tons, largely due to emissions from the primary sector, which had a 46% share. In these emissions, the agriculture subsector contributes 27% over total, and the forestry and the rest of the primary sector, including cattle raising, contributes 19%. Followed by the transportation and manufacturing, industry, and construction sectors with shares of approximately 13% and 12%, respectively. Regarding the emissions of the bioeconomy, these represent 47% and 45% of the total emissions in Colombia and Antioquia, respectively. While by excluding the primary sector from the analysis, it is evident that the contribution of the bioeconomy to total emissions for Colombia and Antioquia becomes 3.6% and 4.1%, respectively. The sector with the highest level of emissions is the manufacturing sector, which represents 88% for Colombia and 90% for Antioquia.

Table 4 relates the GHG ratios between the value added of different groups of economic activity and the calculated bio-shares, with which the contribution of the bioeconomy to total emissions for Colombia is estimated, and Antioquia. As the greater intensity of the ratio of emissions and value added is observed in the forestry sector (it is including other subsectors of the primary sector other than agriculture). This result is in line with the analysis by the Food and Agriculture Organization (FAO) [50], who affirm that livestock threatens the environment, stating that it generates more GHG than the transportation sector, in addition to its impact on the degradation of soil and water resources. In this regard, it is worth to mention that the European Commission is working on a project called "pilot of carbon farming" consisting in establishing a fund for farmers and foresters in order to increase carbon sequestration from soil and biomass and in that way reducing emissions of GHG from livestock and the intensive use of fertilizers in agriculture [51].

**Table 4.** Green House Gas (GHG) intensity in terms of value added and bio-shares by economic activity groups, Colombia.

| Sector | Intensity Coefficient | Bio-Shares |
|---|---|---|
| Agriculture | 1.432 | 0.799 |
| Forestal resto primario | 2.268 | 0.721 |
| Waste | 1.841 | 0.018 |
| Manufacture + industrial processes + Construction | 0.157 | 0.161 |
| Electricity/Heat | 0.797 | 0.007 |
| Transportation | 0.649 | 0.017 |
| Rest of the economy | 0.053 | 0.038 |

Source: Own calculations. Note: GHG emissions are measured in thousands of tons. Intensity is calculated as $\frac{GHG\ emissions}{VA}$.

Figure 5 shows the total and the bioeconomy contribution to GHG emissions by groups of economic activity for Colombia (Figure 5a) and Antioquia (Figure 5b). The general trends are similar for all economic activity groups, where the agricultural and forestry and others represent the highest number of emissions for the bioeconomy at the national and regional level. As it has been recognized that GHG emissions are the main cause of climate change, in this line, the bioeconomy from a sustainable approach offers opportunities to address adaptation to climate change and mitigation of GHG emissions [52]. Furthermore, the quantification in terms of reduction in GHG emissions remains a challenge [53], and policies supporting GHG emissions savings may provide perverse incentives [54].

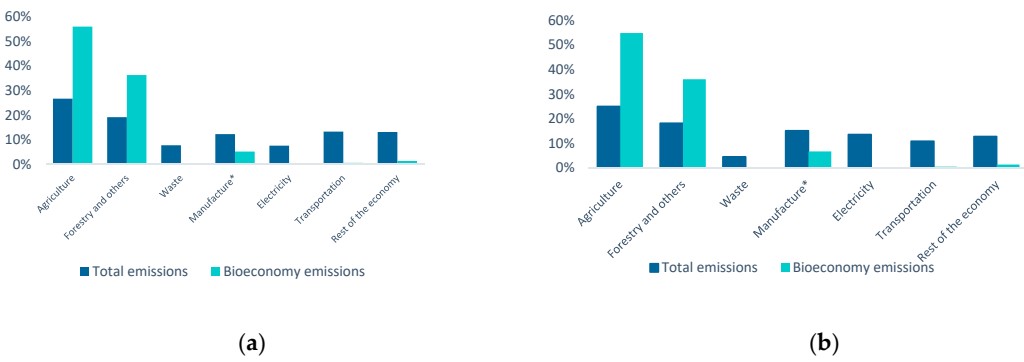

(**a**)                                               (**b**)

**Figure 5.** Bioeconomy and total economic contribution of GHG emissions by groups of economic activities, Source: Own calculations. * Note: Manufacture includes manufacture, industry, and construction sectors. (**a**) Colombia; (**b**) Antioquia.

For its part, the transformation of the industry towards recovery processes, the use of bioenergy generated from biological resources, such as wood, manure, food waste, and algae, allows recycling of GHGs for the generation of energy in the form of fuel and electricity without increasing emissions [55]. This can even have a positive effect on emissions, i.e., it may reduce emissions due to differences in timing between the sequestration and release of $CO_2$ [56]. Thus, regions that have a high share of agriculture and other sectors of the primary sector in their generation of added value. In this sense, it is important to recognize that, although the bioeconomy can enhance the capacities of the territories, exploit their idiosyncratic attributes, and promote their socioeconomic development, research and investment in sustainable processes and activities is necessary, from the social and environmental perspectives.

### 4.4. Indicator 4: Enrollment in Bioeconomy Related Academic Programs

Estimating the share of students enrolled in bioeconomy related academic areas, we found that only 9.8% and 9.2% of total enrollment belongs to bachelor and technical programs in Colombia and Antioquia, respectively. In the case of graduate programs,

the participation is 7.7% in Colombia and 10.1% in Antioquia. Regarding the programs related to bioeconomy, Table 5 shows that environmental, sanitary, biomedical, and related engineering represent 4% of the total enrollments at the professional level for both Colombia and Antioquia. In order, this is followed by the agronomy, agricultural, forestry and related engineering at national level and biology, microbiology chemistry and related at the regional level.

**Table 5.** Participation of bio-related higher education programs in total enrollment by field of knowledge.

| Programs | Bachelor Level | | Graduate Programs | |
|---|---|---|---|---|
| | Colombia | Antioquia | Colombia | Antioquia |
| Agronomy, agricultural, forestry, agronomic, and related engineering | 1.59% | 1.08% | 0.69% | 0.62% |
| Agro-industrial engineering, food, and related | 0.61% | 0.38% | 0.24% | 0.54% |
| Veterinary medicine and zootechnics | 1.19% | 1.80% | 0.31% | 0.46% |
| Environmental, sanitary, biomedical, chemical, and related engineering | 3.96% | 4.02% | 1.99% | 1.78% |
| Biology, microbiology, chemistry, and related | 1.25% | 1.36% | 1.33% | 1.82% |
| Bacteriology and public health | 1.21% | 0.54% | 3.13% | 4.84% |
| Total of bio-related | 9.81% | 9.17% | 7.69% | 10.06% |
| Total of non-bio-related | 90.19% | 90.83% | 92.31% | 89.94% |

Source: Own calculations.

These results mark important challenges from the policy perspective. By increasing the human capital, there is a positive effect on innovation, competitiveness, economic development, and economic growth [57]. However, the geographical distribution of the enrollment in academic programs between larger cities and rural areas (data not shown) implies a challenge for public policies related to education and human capital improvement needed for the bioeconomy. For example, in the case of Antioquia, 96% of the students enrolled in academic programs related to the bioeconomy are located in the metropolitan area. This situation is common in other regions of Colombia and brings several questions about the higher education system since young people in rural areas, where the natural capital including biodiversity is located, have very few educational and training opportunities to be part of the bioeconomy activities. Even in European countries, there is a gap between the human capital available for the bioeconomy sectors, particularly agriculture, and the rest of the economic activities. In fact, the EU agricultural sector shows a lower level of training and education than other economic sectors. In 2016, approximately 40% of workers in the agricultural activities had at most completed a low level of education comparing to 18% of the total working population [51].

Colombia must improve the opportunities for young people to be enrolled in higher education programs related to the bioeconomy. The current enrollment rate is low compared to the potential of natural capital and biological resources to develop the bioeconomy activities, particularly in rural areas. It is precisely in rural areas where the lack of higher education programs is more severe.

### 4.5. Indicator 5: Research Capabilities in Bioeconomy Related Fields

Considering the available data of RG, we found a total of 5770 in Colombia, from which 830 correspond to the region of Antioquia. In terms of their potential contribution to the bioeconomy, we identified only six bioeconomy-related research groups (BRG) in Colombia that explicitly incorporates the term "bioeconomy" within their research lines (Table 6, Search I). From the broader search with the terms from Table 2, we identified 1355 (23.5%) and 220 (26.5%) bioeconomy-related research groups (BRG) in Colombia and Antioquia, respectively. These results show larger research capabilities in the country than those reported by Betancur et al. [58], who identified 618 research groups in Colombia that could contribute to the bioeconomy. Our findings are, on the other hand, in line with the technical report from the National Planning Department (DNP in Spanish), which served to orientate the Green Growth Policy in Colombia [59]. In that report, around 1,500 BRG were identified. It is worthy to mention, however, that the search criteria for the RG is not mentioned in these previous works, and our methodology allows to infer from a deeper analysis of the research field beyond the category that the RG belongs.

**Table 6.** Number of bioeconomy-related research groups (BRG) in Colombia.

| Region | Search I (TOTAL BRG = 6) | | | Search II (TOTAL BRG = 379) | | | Search III (TOTAL BRG = 1355) | | |
|---|---|---|---|---|---|---|---|---|---|
| | Number of BRG | % Over BRG | % Over RG * | Number of BRG | % Over BRG | % Over RG* | Number of BRG | % Over BRG | % Over RG * |
| Antioquia | 2 | 33.3% | 0.03% | 66 | 17.4% | 1.14% | 220 | 16.24% | 3.95% |
| Bogotá | 1 | 16.7% | 0.02% | 115 | 30.3% | 1.99% | 362 | 26.72% | 6.50% |
| Valle del Cauca | 0 | 0% | 0.00% | 31 | 8.2% | 0.54% | 109 | 8.04% | 1.96% |
| Rest of Colombia | 3 | 50% | 0.05% | 167 | 44.1% | 2.89% | 664 | 49.00% | 11.92% |
| TOTAL | 6 | 100% | 0.1% | 379 | 100% | 6.57% | 1355 | 100% | 23.48% |

Source: Own calculations. * Note: Total RG in Colombia = 5770.

As for the distribution among the National Programs of Science and Technology, the research areas more relevant for the advancement of the bioeconomy in Colombia and Antioquia are Agricultural Science; Basic Science; Environment, Biodiversity and Habitat; and Health Science (Table 7). Although the research from Biotechnology program does not stand out yet as one of the main contributors to the bioeconomy, León-de la O et al. reported an increase of 400% in the publication of health biotechnology-related articles from Colombia in the period 2001–2015, coupled with substantial international cooperation in the field [60]. This fact reflects the efforts made by the Colombian authorities to strengthen the biotechnological sector in the country, and, therefore, the advancement of the knowledge-based bioeconomy, supported by several policies, laws, and plans incorporated since 1991 [12].

**Table 7.** Bioeconomy-related research groups by National Program of Science and Technology.

| National Programs of Science and Technology (NPST) | Colombia | | Antioquia | |
|---|---|---|---|---|
| | BRG | % | BRG | % |
| Environment, Biodiversity, and Habitat | 209 | 15.4% | 30 | 13.6% |
| Biotechnology | 85 | 6.3% | 23 | 10.5% |
| Agricultural Science | 262 | 19.3% | 27 | 12.3% |
| Basic Science | 237 | 17.5% | 37 | 16.8% |
| Social Science, Humanities, and Education | 74 | 5.5% | 9 | 4.1% |
| Geosciences | 33 | 2.4% | 0 | 0.0% |
| Energy and Mining | 1 | 0.1% | 5 | 2.3% |
| Engineering | 124 | 9.2% | 26 | 11.8% |
| Uncategorized | 59 | 4.4% | 9 | 4.1% |
| Water Resources | 34 | 2.5% | 2 | 0.9% |
| Health Science | 189 | 13.9% | 51 | 23.2% |
| Security and Defense | 6 | 0.4% | 0 | 0.0% |
| Information and Communication Technology | 42 | 3.1% | 1 | 0.5% |
| TOTAL BGR | 1355 | 100% | 220 | 100% |

Source: Own elaboration.

The research activities in Colombia undergo a process of territorial centralization in the larger cities, and, hence, disregard opportunities for scientific advancement in regions with a larger share in natural resources and biodiversity. Only Bogotá (Colombia's capital) and the departments of Antioquia and Valle del Cauca, accounts for the 51% of the BRG in the country (Figure 6), and in the case of Antioquia and Valle del Cauca, 90% and 65% of BRG correspond to Medellín and Cali, their respective capital cities. This could be explained by the demographic concentration and the larger number of universities installed in these regions, since they are regarded as knowledge producers and training providers [60]. On the other hand, regions such as Amazonas, Chocó, Caquetá, and Meta, four of the most biodiverse areas, which have larger forest cover, but less universities (Appendix A, Figure A2), account for the 4.4.% of the BRG in Colombia.

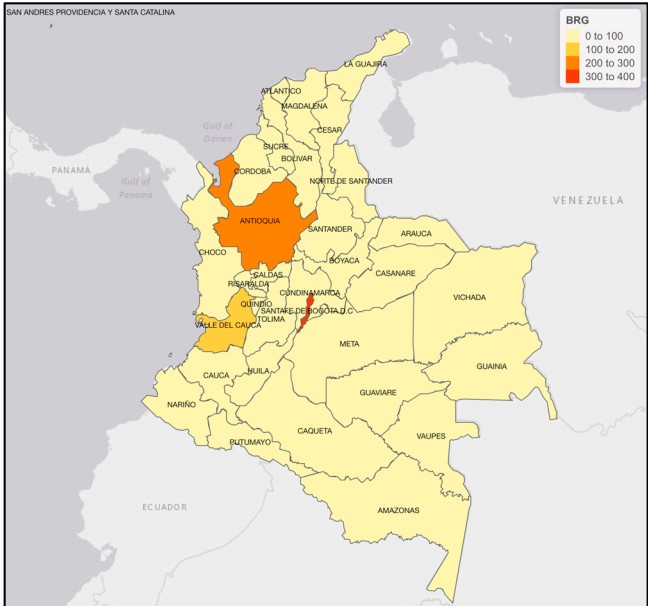

**Figure 6.** Bioeconomy-related research groups distribution in Colombia. Source: own elaboration.

Finally, to have some insights about the trends in research topics of the identified BRG, we study the most recurrent word in the title publications. This enables to infer the research interests in a broad sense (Appendix A, Figure A2, panels a–k). There is

special focus on the processes of characterization and production. This applies to crops of commercial interests such as coffee, potato, and nightshade plants, and forests with potential application in sustainable silviculture. It also applies to cattle, milk, and dairy products. It is also recurrent in the taxonomic classification of species, particularly of insects. The energetic production based on biofuels and biomass use stands out in the mining and energy sector, whereas the study of business and education are highlighted from the Social Science NPST. From the Health Sciences and Biotechnology NPSTs, there is interest in the study of cancer and several viral infections, as well as in molecular biology and diagnostics. Notably, the Caribbean has received more attention in the research conducted by the BRG than the Pacific Ocean, which highlights both a necessity and the opportunity to explore and study the territories in the western region of Colombia.

Moreover, considering not only the number of BRG, but what percentage they represent with respect of the total amount of RG in each region, we can identify the intrinsic relationship with the bioeconomy on the research conducted in some departments, despite having less RG (Figure 7). Those departments that report RG and belong to the natural region of Orinoquía (Orinoco River watershed, near to Venezuela) such as Arauca, Casanare, Meta, and Vichada; to the Amazon such as Amazonas, Caquetá, and Guaviare; and to the Pacific region, particularly Chocó, are more oriented towards bioeconomy-related research. This means that they have a larger share of their RG devoted to bioeconomy-related research, but less RG in general.

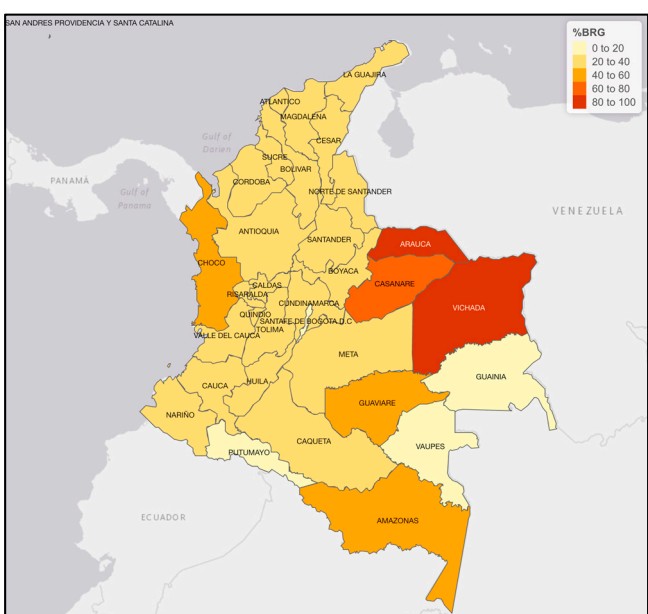

**Figure 7.** Percentage of BRG with respect of the total RG in each region of Colombia. Source: own elaboration.

There have been initiatives such as Colombia Bio (https://minciencias.gov.co/portafolio/colombia-bio, last accessed on 2 November 2020) promoted by the Colombian government to explore the biodiversity, its conservation, and sustainable management through science, technology, and innovation. However, unless there is a strong commitment to expand the academic offers relevant for the bioeconomy, both at the undergraduate and graduate levels, and to increase and bolster the installed capacities to conduct high-impact research within the territories, it is unlikely to develop a strong knowledge-based bioeconomy.

Most bioeconomy policies around the world do not consider the diffusion and use of bioeconomy-related knowledge within the productive system, nor the different types of knowledges that intervene for a sustainable bioeconomy framework. The focus has traditionally been on R&D expenditures to generate the necessary knowledge to create economic value, and this not necessarily implies the achievement of sustainable goals [39].

Particularly, in the case of the Colombian National Strategy of Green Growth, technological transfer is mentioned, but a detailed description of the mechanisms to do so is not provided.

Our analysis on research tendencies is limited to a set of words related to the bioeconomy over the publication titles reported in the GrupLAC, and not the words contained in research papers and publication per se. However, the results are sufficient to picture the relationship between the research conducted in Colombia and the potential productive pathways of the bioeconomy, as proposed by Trigo et al. (2013) [61]. In general terms, the productive pathways that had more research associated with them, corresponded to eco-intensification, ecosystem services, and leveraging on the biodiversity resources. Our approach addressed in a general way the "creation" of knowledge. However, given the cultural and biological diversity among the regions of Colombia, the diffusion and implementation of knowledge (systems, normative, and transformative) should be further analyzed at a regional and local level to describe how the knowledge impacts the bioeconomy in a more detailed fashion. To our knowledge, this is the first work that maps the installed capacity in Colombia to conduct bioeconomy-related research in terms of research groups, and that evaluates the knowledge generation relevant to the bioeconomy using text-mining techniques.

*4.6. Indicators of the Bioeconomy in Colombia and Antioquia*

Table 8 presents a wrap-up of the five proposed indicators to monitor the bioeconomy in Colombia and Antioquia. Updating these estimates provide a benchmark to assess the implementation of public policy towards a sustainable bioeconomy.

**Table 8.** Summary of the bioeconomy indicators for Colombia and Antioquia.

| Indicator | Colombia | Antioquia |
|:---:|:---:|:---:|
| **% VA** | 13.7% | 11.2% |
| **% Employment** | 21.1% | 16.3% |
| **% GHG** | 47% | 45% |
| **%Enrollment in academic programs– Bachelor/Graduate level** | 9.8%/7.7% | 9.2%/10.1% |
| **% BRG** | 23.5% | 26.5% |

Source: Own calculations

## 5. Concluding Remarks

The introduction of bioeconomy is considered a strong approach to act against current trends of unsustainability. Policy makers, academia, and private organizations are shaping alternative models to tackle the impacts of climate change and boost a more sustainable world from social, economic, and environmental perspectives. Fostering bioeconomy has gained an important space in the debate and draws new work lines for the transition towards a greening economy. Cleaner productive process, but also developing bio-based products based on knowledge and technology are the basis of this transition process, where biodiversity is a crucial input, particularly for those countries having abundant biological resources such as the tropical ones.

This study underlines the lack of a unified framework to analyze the contribution of the bioeconomy across countries, which does not allow any straightforward comparison of the relevance of bioeconomy among different economies. The measurement of the bioeconomy becomes a policy input, understanding that it reveals the capacities of the territories to generate value added in different sectors of the economy with the use of natural renewable resources and goods provided by the primary sector as it was defined above. In this way, with the calculation of the size of the bioeconomy in Colombia and Antioquia, efforts could be directed towards the comparative advantages of its local bioeconomies, fostering the development of sustainable practices within all sectors. However, additional effort is

needed to consider social and environmental dimensions, but our results are the baseline for monitoring the bioeconomy in the case of Colombia and Antioquia.

When measuring the bioeconomy in Colombia, the highest participation is attributed to the primary sector with approximately 52%. Overall, the potential of the Colombian economy for sustainable development of the biobased sectors is evident, with the aim of obtaining greater profitability, but above all, achieving sustainability of the resource base in the territories including people living in rural areas. In other words, such potential must be converted into employment, income increase, and quality of life for its inhabitants, in one expression, sustainable rural development. The results suggest that the contribution of the bioeconomy varies importantly across countries. For instance, in the case of EU-28, Kuosmanen et al. [10] found that it could range from 5% to 15% of the value added, while estimation about the employment in bio-based economic activities is around 15%. We contribute to this literature by studying the contribution of the bioeconomy in Colombia in the valued added and the total employment and extend the analysis to the case of the region of Antioquia to provide useful information on the size of bioeconomy and identify opportunities at local economies level. Antioquia is an important region of Colombia not only for its biodiversity richness but also for its contribution to the national GDP. Colombia and Antioquia are interesting cases of study because of their rich natural capital and the pronounced regional disparities that make it important to think the development from the potential of the local bioeconomies. The proposed approach allows in principle to also calculate the bioeconomy contribution to GHG emissions. However, it does not consider carbon-sinks such as forestry and implies the debatable assumption that bio-based activities in hybrid sectors emit the same amount of GHG as their fossil counterpart. The primary production (i.e., agriculture) appears as the major contributor to GHG emissions.

Regarding education in areas related to bioeconomy, we found that there is a lack of students enrolled in academic programs associated with the bioeconomy activities. In the case of Colombia, less than 10% of total students enrolled in higher education are in programs related to bioeconomy such as biology, biotechnology, environmental engineering, agricultural sciences, livestock sciences, health sciences, natural sciences, among others. With respect to local capabilities to generate relevant research in bioeconomy, we observe that the centralization is evident in the process of generating scientific knowledge relevant to the bioeconomy mainly within Bogotá, Antioquia, and Valle del Cauca (51% BRG), whereas territories rich in bioresources have less RG, but are generally more oriented towards bioeconomy-related research. Given the importance of bioeconomy as new engine of green economic development, this result reveals the need of fostering the science and technology ecosystem addressed to take advantage of the local resources.

These results remark the importance of monitoring the bioeconomy for defining priorities of public policy. Colombia is one of the Latin American countries where bioeconomy has called the attention of policy makers. Perhaps one of the key milestones in terms of the sustainable development perspective in Colombia has been the national Constitution in 1991 that established the basis to create the national environmental system in order to promote policies and strategies to use and protect the natural resources base in a sustainable way. From there, national biodiversity policy and the most recent green growth policy have laid the foundations of the, in construction, national bioeconomy strategy [62]. Our analysis might provide important insights on the elements that a regional bioeconomy strategy should have to complement this national strategy.

For developing countries, and especially for tropical countries like Colombia, the second richest country in biodiversity in the world, after Brazil, the bioeconomy appears to be an economic alternative that allows generating value, developing markets, and promoting economic and social growth at national and local level. Indeed, regional, and local economies are important places in the bioeconomy strategies as biodiversity is usually located in rural areas. Developing local innovation ecosystems, infrastructure, and facilities, as well as local markets connected to the whole supply chain, are factors to successfully implement a bioeconomy strategy [24,63,64].

**Author Contributions:** Data curation, A.G.-S., L.R.-G. and S.V.-V.; Formal analysis, A.G.-S.; Investigation, M.A.; Methodology, A.G.-S.; Project administration, M.A.; Software, L.R.-G. and S.V.-V.; Visualization, L.R.-G. and S.V.-V.; Writing – original draft, M.A., A.G.-S., L.R.-G. and S.V.-V. All authors have read and agreed to the published version of the manuscript.

**Funding:** This research was funded by research grants of Universidad EIA. This research was also supported by the Colombia Científica-Alianza EFI Research Program, with code 60185 and contract number FP44842-220-2018, funded by The World Bank through the call Scientific Ecosystems, managed by the Colombian Ministry of Science, Technology and Innovation.

**Institutional Review Board Statement:** Not applicable.

**Informed Consent Statement:** Not applicable.

**Data Availability Statement:** The data presented in this study are available on request from the corresponding author.

**Acknowledgments:** We thank the comments of the attendees at the Conference of the International Consortium on Applied Bioeconomy Research (ICABR) 2020, and comments and suggestions by Nathalia Flórez-Zapata and Jose Pablo Navarro. Errors, opinions, and omissions are ours and do not compromise the institutions.

**Conflicts of Interest:** The authors declare no conflict of interest.

**Appendix A**

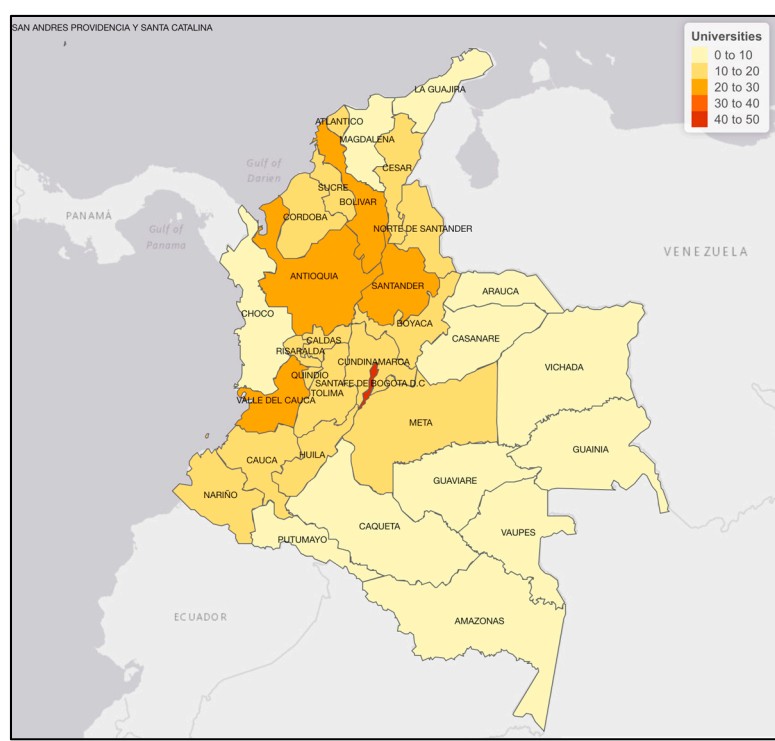

**Figure A1.** Universities distribution in Colombia. Source: Own calculations.

## a. Environment, Biodiversity and Habitat NPST. Total articles: 13,699

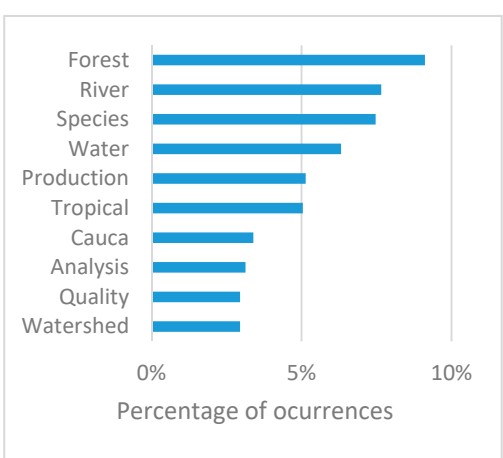

## b. Biotechnology NPST. Total articles: 6192. (*) form Spanish we were not able to identify differences.

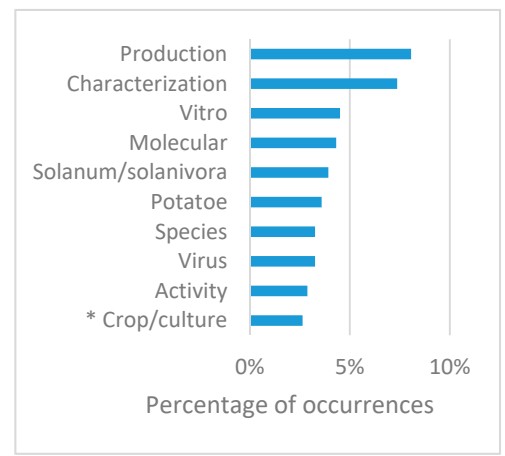

## c. Agricultural Sciences NPST. Total articles: 24,949

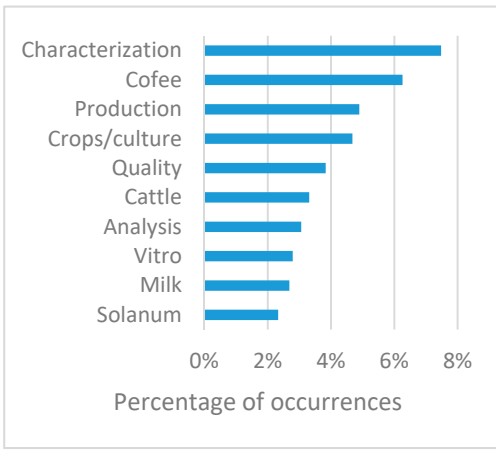

## d. Basic Science NPST. Total articles: 18,511. (*) Different taxonomical categories.

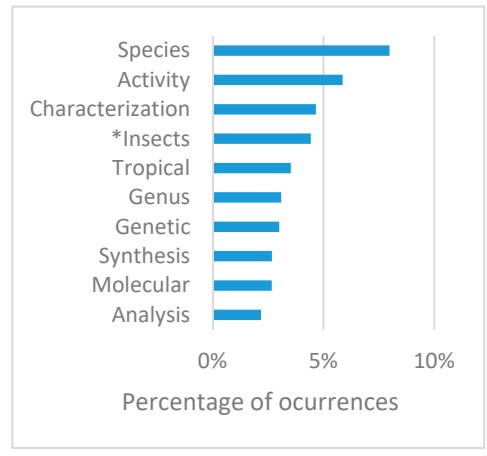

## e. Social Science, Humanities and Education NPST. Total articles: 2990

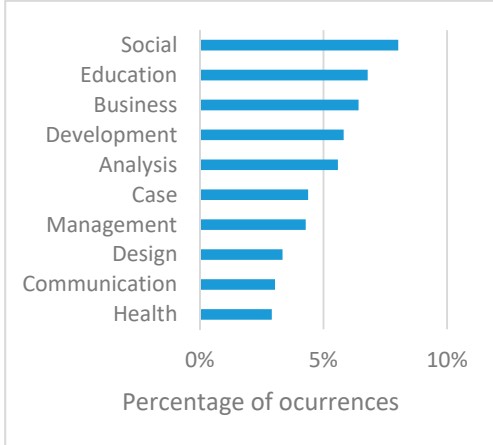

## f. Energy and Mining NPST. Total articles: 2858

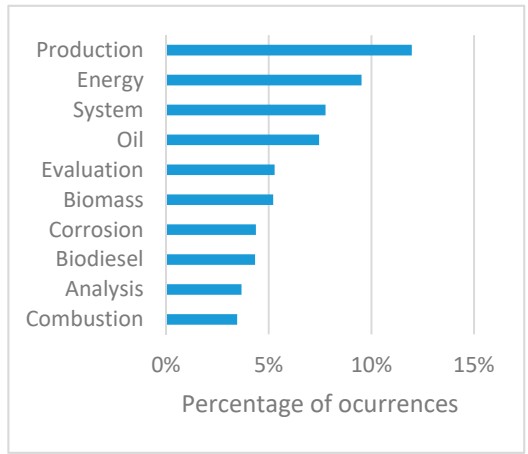

**Figure A2.** *Cont.*

**g. Engineering NPST. Total articles: 7017**

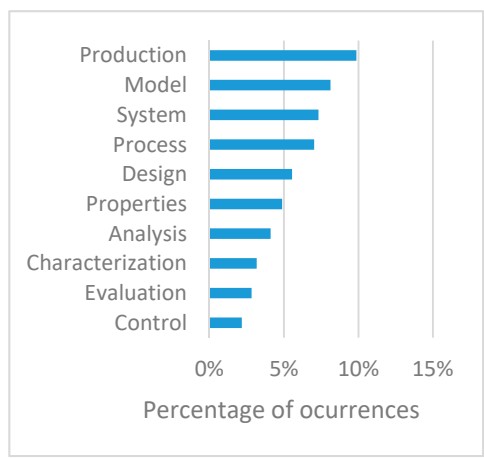

**h. Uncategorized NPST. Total articles: 2266**

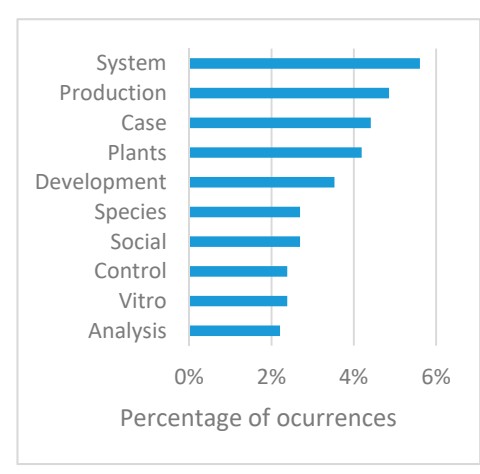

**i. Water Resources NPST. Total articles: 3166**

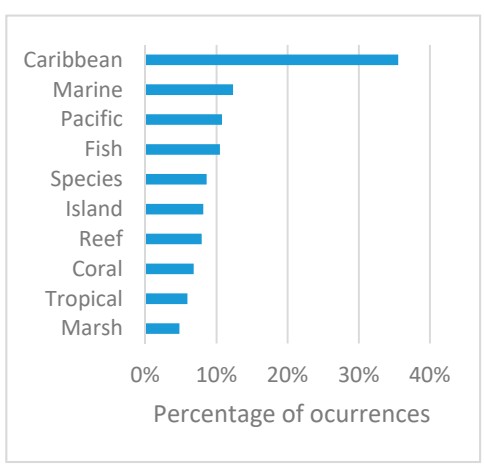

**j. Health Sciences NPST. Total articles: 23,015**

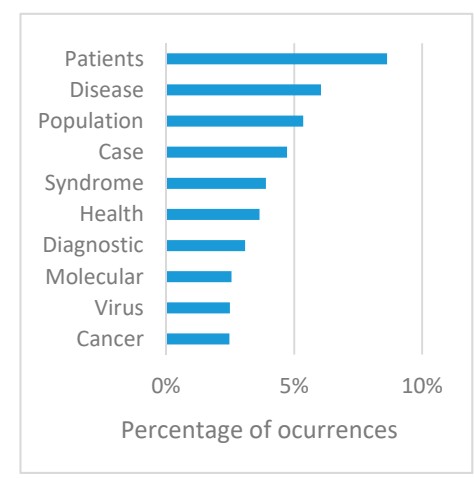

**k. ICT NPST. Total articles: 2771**

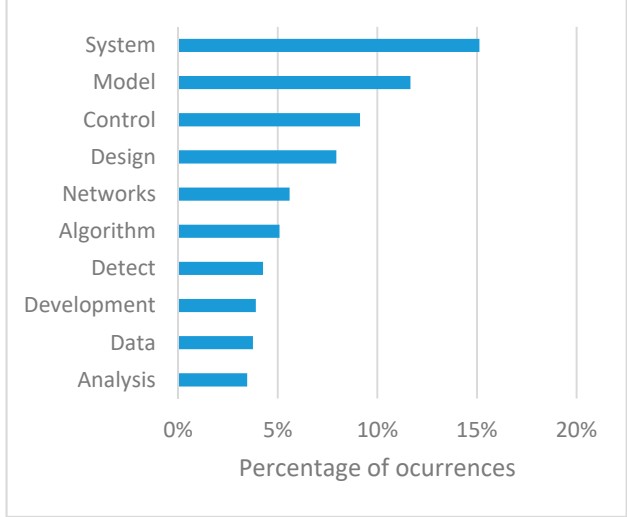

**Figure A2.** Top 10 words from academic production of bioeconomy-related research groups (BRG) in Colombia from the different National Programs of Science and Technology (NPST). Source: Own calculations.

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
