# Peer review of "Measuring the Contribution of the Bioeconomy: The Case of Colombia and Antioquia"

_sustainability, doi:10.3390/su13042353_

Round 1

Reviewer 1 Report

The authors improved considerably the manuscript according with the revisers' comments and suggestions. 

Author Response

Dear reviewer,

thanks for your comments. We have reviewed our paper and made changes (using the "Track Changes"), mainly typographical errors. We have also shortened some paragraphs that seemed to be repetitive. This includes a shorter version of the abstract.

We wish to reiterate our gratitude and hope this new version of the paper reflects your suggestions.

Kind regards

Reviewer 2 Report

In the introduction, the authors presented the global conditions of the research problem, they also described the concept of bioeconomy and gave its definitions. The contribution of the bioeconomy to creating added value was then identified. The positioning of the research topic in theory is correct. In the revised version of the article, the authors added the purpose of the work, which I assess positively. The authors also made a justification for the selection of Colombia and the Antioquia region for research.

The Methods and data section has been added. This part is well described, supplemented with patterns.

In the Results and discussion section, the authors present the contribution of the bioeconomy in Colombia and Antioquia. The authors comment on the results achieved on an ongoing basis. The division into subsections is correct. Summary is too long. It should be more synthetic and compact. Authors have a tendency to write voluminously.

There were errors in the links to the tables and figures that were in the text. Example (pages 5, 9-19): see Error! Reference source not found

The authors made changes to the article compared to the previous version. Now the article is more compact, it contains the necessary parts. Literature has been replenished. Its number of items is acceptable as it stands.

Author Response

Dear reviewer,

thanks for your comments. We have reviewed our paper and made changes (using the "Track Changes"), mainly typographical errors. We have also shortened some paragraphs (including the summary in Section 5) that seemed to be repetitive. This includes a shorter version of the abstract.

Error in the cross-referencing and numbering were double checked.

We wish to reiterate our gratitude and hope this new version of the paper reflects your suggestions.

Kind regards

This manuscript is a resubmission of an earlier submission. The following is a list of the peer review reports and author responses from that submission.

Round 1

Reviewer 1 Report

This manuscript provides evidence of the contribution of the bioeconomy to the value added and employment inColombia. As a robustness test, it also analyses the bioeconomy in one region in Colombia, Antioquia. The methodology is based on the analysis of input-output matrices, calculating the bioshares in the output. Results show that 11.03% of the gross value added in Colombia is due to 126 activities associated to the bioeconomy, which also corresponds to the 21.10% of the total 127 employment. For the Antioquia region, this percentages is 11.08% and 16.74% respectively.

This manuscript is very interesting and show relevant results and policy implications.

Comments and suggestions:

The manuscript needs to be restructured and rewritten.

An English proof revision is also required.

The abstract should mention the methodology used. Also, it should be written in a more appealing way.

The introduction is too long. It should be rewritten in a concise and direct manner, and the paragraphs that review the literature should be gathered in a section of related literature. Example paragraph from line 89 is more of related literature than to be placed in an introduction.

In section 2, first paragraphs, From line 169, it is very confusing.

In page 5, the equations should be numbered.

In line 196, the Variable D should be explained before the equation, otherwise, it is very difficult to understand the equation.

In the paragraph that starts in line 235, relative to the figures (number) presented the authors should explained how were calculated. Again, explain how the figures in paragraph from line 246 were obtained?

A subsection in line 259, should be created to introduce the analysis of the employment.

The paragraph from line 295 is very confusing.

From line 305, a subsection related to the analysis of Antioquia.

Section 3 is too long and confusing. It should be rewritten and restructured.

After the discussion, a section with the conclusion should be presented.

Minor comments

Line 78 row materials, replace raw materials

Line 91 What is JRC? – please before using acronyms write it in extent.

Line 97 that that?

Line 112 too the case – to the case

Line 322 approach than in the indicator to measure?

In table 7, Meddl n?

Line 519 remove enter

Line 526, remove enter

Line 551, remove enter.

Reviewer 2 Report

The authors in the Introduction described the concept of bioeconomics. They gave her definitions. Then, the contribution of bioeconomics to creating added value was determined. It varied from country to country. The next step presents the contribution of bioeconomy in Colombia and Antioquia.
The article has a number of disadvantages. My comments are as follows.
The layout of the work is generally inadequate. There is no separate part concerning the research methodology, as well as separate goals and hypotheses. There is no date when the tests were performed, no method of selecting test objects, no description of the research methods used. There are no very advanced methods of analysis. Doubts are raised by the selection of areas for research, the entire country and one of its regions. Why is this region so special compared to others? There should be such a comparison with other regions.
There is no discussion relating to the results obtained in the article. What results did others get, are they similar or not? The discussion in the article actually relates to the Summary.

Reviewer 3 Report

The first section of the article attempts to clarify the concept of the bioeconomy and to show its growing success in various political and socio-economic arenas.

It aims to analyze the contributions of the bioeconomy at the scale of Colombia and the Antioquia region, through various measurement methods.

More specifically, it aims to estimate the contribution of the bio-based economy to the whole economy. The sectors of the economy addressed are interesting, including large value added, employment, education, research capabilities, etc.

It is unfortunate that the data are not further contextualized, analyzed and critiqued and that the method does not address all dimensions of sustainability (ecological, economic and social), which weakens the scope of this paper and its contribution to the topic.